# Feed-forward Propagation in Probabilistic Neural Networks with Categorical and Max Layers

**Alexander Shekhovtsov & Boris Flach**
Department of Cybernectics, Czech Technical University in Prague
Zikova 4, 166 36 Prague
{shekhole,flachbor}@fel.cvut.cz

## Abstract

Probabilistic Neural Networks deal with various sources of stochasticity: input noise, dropout, stochastic neurons, parameter uncertainties modeled as random variables, etc. In this paper we revisit a feed-forward propagation approach that allows one to estimate for each neuron its mean and variance w.r.t. all mentioned sources of stochasticity. In contrast, standard NNs propagate only point estimates, discarding the uncertainty. Methods propagating also the variance have been proposed by several authors in different context. The view presented here attempts to clarify the assumptions and derivation behind such methods, relate them to classical NNs and broaden their scope of applicability. The main technical contributions are new approximations for the distributions of argmax and max-related transforms, which allow for fully analytic uncertainty propagation in networks with softmax and max-pooling layers as well as leaky ReLU activations. We evaluate the accuracy of the approximation and suggest a simple calibration. Applying the method to networks with dropout allows for faster training and gives improved test likelihoods without the need of sampling.

## 1 Introduction

Despite the massive success of Neural Networks (NNs) considered as deterministic predictors, there are many scenarios where a probabilistic treatment is highly desirable. One of the best known techniques to improve the network generalization is *dropout* (Srivastava et al., 2014), which introduces multiplicative Bernoulli noise in the network. At test time, however, it is commonly approximated by substituting the mean value of the noise variables. Computing the expectation more accurately by Monte Carlo (MC) sampling has been shown to improve test likelihood and accuracy (Srivastava et al., 2014; Gal & Ghahramani, 2015) but is computationally expensive.

Another challenging problem in NNs is the sensitivity of the output to perturbations of the input, in particular random and adversarial perturbations (Moosavi-Dezfooli et al., 2017; Fawzi et al., 2016; Rodner et al., 2016). In Fig. 1 we illustrate the point that the average of the network output under noisy input differs from propagating the clean input. It is therefore desirable to estimate the output uncertainty resulting from the uncertainty of the input. In classification networks, propagating the uncertainty of the input can impact the confidence of the classifier and its robustness (Astudillo & da Silva Neto, 2011). We would like that a classifier is not overconfident when making errors. However such high confidences of wrong predictions are typically observed in NNs. Similarly, when predicting real values (*e.g.* in optical flow estimation), it is desirable to estimate also their confidences. Taking into account uncertainties from input or dropout allows to predict output uncertainties better correlated with the test error (Kendall & Gal, 2017; Gast & Roth, 2018; Schoenholz et al., 2016). Another important problem is overfitting, which may be addressed in a sound way with Bayesian learning. The parameters are considered as random variables and are determined up to an uncertainty implied by the training data. This uncertainty needs then to be propagated to predictions at the test-time.

The above scenarios motivate considering NNs with different sources of stochasticity not as deterministic feed-forward networks but as directed probabilistic graphical models. We focus on the

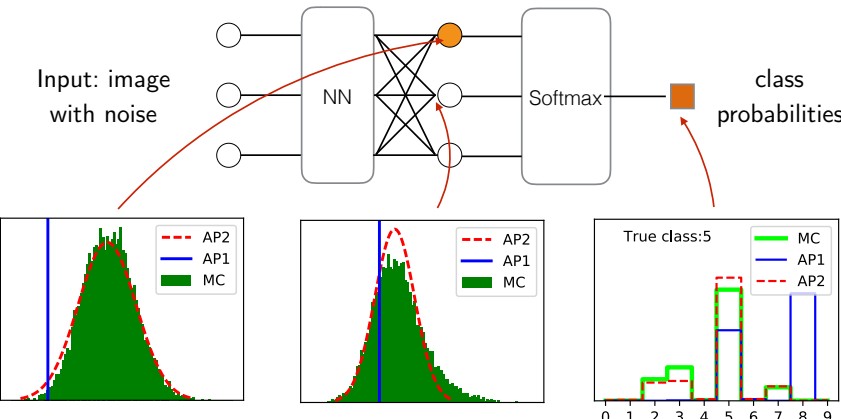

Figure 1: Propagating an input perturbed with Gaussian noise $\mathcal{N}(0, 0.1)$ through a fully trained LeNet. When the same image is perturbed with different noise samples, we observe in the hidden units and on the output an empirical distributions shown as Monte Carlo (MC) histograms. Propagating the clean image results in the estimate denoted AP1 which may be away from the MC mean. Propagating means and variances results in a posterior Gaussian distribution denoted AP2. For the output class probabilities we need to approximate the expected value of the softmax. The methods AP1 and AP2 are formally defined in § 3 and a quantitative evaluation will be given in § 5.

*inference problem* that consists in estimating the probability of hidden units and the outputs given the network input, referred to as *posterior* distributions. While there exist elaborate inference methods such as variational, belief propagation, Gibbs sampling, *etc.*, they are computationally demanding and can hardly be applied at the same scale as state-of-the-art NNs.

**Contribution and Related Work** We revisit feed-forward propagation methods that perform an approximate inference analytically by propagating means and variances of neurons through all layers of a NN, ensuring computational efficiency and differentiability. This type of propagation has been proposed by several authors under different names: *uncertainty propagation* (Astudillo & da Silva Neto, 2011; Astudillo et al., 2014), *fast dropout training* (Wang & Manning, 2013), *probabilistic backpropagation* (Hernández-Lobato & Adams, 2015) in the context of Bayesian learning, *assumed density filtering* (Gast & Roth, 2018). Perhaps the most general form is considered by Wang et al. (2016) and termed *natural parameter networks*. The *local reparametrization trick* (Kingma et al., 2015) can be viewed as an application of the variance propagation method through one layer only and then sampling from the approximate distribution.

These preceding works were using sampling or point estimates for propagation through softmax and avoided max-pooling. Ghosh et al. (2016) proposed an analytic approximation for softmax but resorted to sampling, noting that the approximation was not accurate. Gast & Roth (2018) introduced Dirichlet posterior to overcome the difficulty with softmax, however, the softmax is still used in the model internally. Astudillo et al. (2014) achieved improvements in speech recognition with the uncertainty propagation but explicitly mentions the approximation for softmax as an unsolved problem and a significant limitation. Lastly, the expressions for ReLU activations that are typically used involve differences of error functions and may be unstable.

We propose a latent variable view of probabilistic NNs, which shows the connection to standard NNs and allows us to develop better approximations. We develop numerically suitable approximations for propagating means and variances through multivariate functions such as $\mathrm{softmax}$, $\mathrm{argmax}$ and log-sum-exp to handle categorical distributions as well as $\max$-related non-linearities: $\max$-pooling and leaky ReLU. This makes the propagation approach applicable to a wider class of problems.

Experimentally, we verify the accuracy of the proposed approximations as well as of the whole propagation method and compare it to the standard NN. This verification shows that our approximations are accurate in comparison with sampling and that the variance propagation method estimates the output distribution of the network significantly better than the standard NN. We further demonstrate its potential utility in the end-to-end learning with dropout.

## 2    PROBABILISTIC NNS AND FEED-FORWARD EXPECTATION PROPAGATION

In probabilistic NNs, all units are considered random. In a typical network, units are organized by layers. There are $l$ layers of hidden random vectors $X^k$, $k = 1, \ldots l$ and $X^0$ is the input layer. Each vector $X^k$ has $n_k$ components (layer units) denoted $X_i^k$. The network is modeled as a conditional *Bayesian network* (a.k.a. belief network, Neal (1992)) defined by the pdf

$$p(X^{1,\ldots l} \mid X^0) = \prod_{k=1}^{l} p(X^k \mid X^{k-1}). \tag{1}$$

We further assume that the conditional distribution $p(X^k \mid X^{k-1})$ factorizes as $p(X^k \mid X^{k-1}) = \prod_{i=1}^{n_k} p(X_i^k \mid A_i^k)$, where $A_i^k = \sum_j w_{ij}^k X_j^{k-1}$ are *activations*. In this work we do not consider Bayesian learning and the weights $w$ are assumed to be non-random, for clarity. We will denote values of r.v. $X^k$ by $x^k$, so that the event $X^k = x^k$ can be unambiguously denoted just by $x^k$. Notice also that we consider biases of the units implicitly via an additional input fixed to value one. The posterior distribution of each layer $k > 0$, given the observations $x^0$, recurrently expresses as

$$p(X^k \mid x^0) = \mathbb{E}_{X^{k-1} \mid x^0}\left[p(X^k \mid X^{k-1})\right] = \int p(X^k \mid x^{k-1})p(x^{k-1} \mid x^0)\, dx^{k-1}. \tag{2}$$

The posterior distribution of the last layer, $p(X^l \mid x^0)$ is the model's predictive distribution.

Standard NNs with injected noises give rise to Bayesian networks of the form (1) as follows. Consider a deterministic nonlinear mapping applied component-wise to *noised* activations:

$$X_i^k = f(A_i^k - Z_i^k), \tag{3}$$

where $f \colon \mathbb{R} \to \mathbb{R}$ and $Z_i^k$ are independent real-valued random variables with a known distribution (such as the standard normal distribution). From representation (3) we can recover the conditional cdf $F_{X_i^k \mid X^{k-1}}(u) = \mathbb{E}[\![f(w_i^{k\mathsf{T}} X_i^{k-1} - Z^k) \leq u \mid X^{k-1}]\!]$ and the respective conditional density of the belief network.

**Example 1.** *Stochastic binary unit (Williams, 1992).* Let $Y$ be a binary valued r.v. given by $Y = \Theta(A - Z)$, where $\Theta$ is the Heaviside step function and $Z$ is noise with cdf $F_Z$. Then $\mathbb{P}(Y{=}1 \mid A) = F_Z(A)$. This is easily seen from

$$\mathbb{P}(Y{=}1 \mid A) = \mathbb{P}(\Theta(A - Z) = 1 \mid A) = \mathbb{P}(Z \leq A \mid A) = F_Z(A). \tag{4}$$

If, for instance, $Z$ has standard logistic distribution, then $\mathbb{P}(Y{=}1 \mid A) = \mathcal{S}(A)$, where $\mathcal{S}$ is the *logistic sigmoid* function $\mathcal{S}(a) = (1 + e^{-a})^{-1}$. $\qquad\square$

In general, the expectation (2) is intractable to compute and the resulting posterior can have a combinatorial number of modes. However, in many cases of interest it is suitable to approximate the posterior $p(X^k \mid x^0)$ for a given $x^0$ with a factorized distribution $q(X^k) = \prod_i q(X_i^k)$. We expect that in many recognition problems, given the input image, all hidden states and the final prediction are concentrated around some specific values (unlike in generative problems, where the posterior distributions are typically multi-modal). A similar factorized approximation is made for the activations. The exact shape of distributions $q(X_i^k)$ and $q(A_i^k)$ can be chosen appropriately depending on the unit type: *e.g.*, a Bernoulli distribution for binary $X_i^k$ a Gaussian or Logistic distribution for real-valued activations $A_i^k$. We will rely on the fact that the mean and variance are sufficient statistics for such approximating distributions. Then, as long as we can calculate these sufficient statistics for the layer of interest, the exact shape of distributions for the intermediate outputs need not be assumed.

The information-theoretic optimal factorized approximation to the posterior $p(X^k \mid x^0)$ minimizes the forward KL divergence $KL(p(X^k \mid x^0) \| q(X^k))$ and is given by the marginals $\prod_i p(X_i^k \mid x^0)$. Furthermore, in the case when $q(X_i^k)$ is from an exponential family, the optimal approximation is given by matching the moments of $q(X_i^k)$ to $p(X_i^k \mid x^0)$. The factorized approximation then can be computed layer-by-layer, assuming that the preceding layer was already approximated. Substituting $q(X^{k-1})$ for $p(X^{k-1} \mid x^0)$ in (2) results in the procedure

$$q(X_i^k) = \mathbb{E}_{q(X^{k-1})}\left[p(X_i^k \mid X^{k-1})\right] = \int p(X_i^k \mid x^{k-1}) \prod_i q(x_i^{k-1})\, dx^{k-1}. \tag{5}$$

Thus we need to propagate the factorized approximation layer-by-layer with the marginalization update (5) until we get the approximate posterior output $q(X^l)$. This method is closely related to the *assumed density filtering* (see Minka, 2001), in which, in the context of learning, one chooses a family of distributions that is easy to work with and "projects" the true posterior onto the family after each measurement update. Here, the projection takes place after propagating each layer, for the purpose of the inference.

## 3 PROPAGATION IN BASIC LAYERS

We now detail how (5) is computed (approximately) for a single layer consisting of a linear mapping $A = w^\mathsf{T} X$ (scalar output, for clarity) and a non-linear noisy activation $Y = f(A - Z)$.

**Linear Mapping** An activation $A$ in a typical deep network is a linear combination of many inputs $X$ from the previous layer. This justifies the assumption that $A - Z$ (where $Z$ is a smoothly distributed injected noise) can be approximated by a uni-modal distribution fully specified by its mean and variance such as normal or logistic distribution[1]. Knowing the statistics of $Z$, we can estimate the mean and the variance of the activation $A$ as

$$\mu' = \mathbb{E}[A] = w^\mathsf{T} \mathbb{E}[X] = w^\mathsf{T}\mu, \tag{6a}$$

$$\sigma'^2 = \sum_{ij} w_i w_j \operatorname{Cov}[X]_{ij} \approx \sum_i w_i^2 \sigma_i^2, \tag{6b}$$

where $\mu$ is the mean and $\operatorname{Cov}[X]$ is the covariance matrix of $X$. The approximation of the covariance matrix by its diagonal is implied by the factorization assumption for the activations $A$.

**Nonlinear Coordinate-wise Mappings** Let $A$ be a scalar r.v. with statistics $(\mu, \sigma^2)$ and let $Y = f(A - Z)$ with independent noise $Z$. Assuming that $\widetilde{A} = A - Z$ is distributed normally or logistically with statistics $\tilde{\mu}, \tilde{\sigma}^2$, we can approximate the expectation and the variance of $Y = f(\widetilde{A})$,

$$\mu'_i = \mathbb{E}_{q(\widetilde{A})}[f(\widetilde{A})], \qquad \sigma'^2_i = \mathbb{E}_{q(\widetilde{A})}[f^2(\widetilde{A})] - \mu'^2_i, \tag{7}$$

by analytic expressions for most of the commonly used non-linearities. For binary variables, occurring in networks with Heaviside nonlinearities, the distribution $q(Y)$ is fully described by one parameter $\mu_i = \mathbb{E}[Y]$, and the propagation rule (5) becomes

$$\mu'_i = \mathbb{E}_{q(A)}\big[p(Y{=}1 \,|\, A^k)\big], \qquad \sigma'^2_i = \mu'_i(1 - \mu'_i), \tag{8}$$

where the variance is dependent but will be needed in propagation through further layers.

**Example 2.** *Heaviside Nonlinearity with Noise.* Consider the model $Y = \Theta(A - Z)$, where $Z$ is logistic noise. The statistics of $\widetilde{A} = A - Z$ are given by $\tilde{\mu} = \mu$ and $\tilde{\sigma}^2 = \sigma^2 + \sigma_S^2$, where $\sigma_S^2 = \pi^2/3$ is the variance of $Z$. Assuming noisy activations $\widetilde{A}$ to have logistic distribution, we obtain the mean of $Y$ as:

$$\mu' = \mathbb{E}[\Theta(\tilde{A})] = \mathbb{P}(\widetilde{A} \geq 0) = \mathbb{P}\Big(\frac{\widetilde{A} - \tilde{\mu}}{\tilde{\sigma}/\sigma_S} \geq \frac{-\tilde{\mu}}{\tilde{\sigma}/\sigma_S}\Big) \doteq \mathcal{S}\Big(\frac{\tilde{\mu}}{\tilde{\sigma}/\sigma_S}\Big) = \mathcal{S}\Big(\frac{\mu}{\sqrt{\sigma^2/\sigma_S^2 + 1}}\Big), \quad (9)$$

where the dotted equality holds because $-(\widetilde{A} - \tilde{\mu})\frac{\sigma_S}{\tilde{\sigma}}$ has standard logistic distribution whose cdf is the sigmoid function $S$. The variance of $Y$ is expressed as in (8).

Fig. 2 shows approximations for propagation through Heaviside function and through (leaky) ReLU detailed in § 4.3. Note that all expectations over a smoothly distributed $A$ result in smooth propagation functions regardless the smoothness (or lack thereof) of the original function.

Summarizing, we can represent the approximate inference in networks with binary and continuous variables as a feed-forward moment propagation: given the approximate moments of $X^{k-1} \,|\, x^0$, the moments of $X_i^k \,|\, x^0$ are estimated via (7)-(8) ignoring dependencies between $X_j^{k-1} \,|\, x^0$ on each step (as implied by the factorized approximation).

---

[1]Note, the prior work assumes that $A$ alone approaches Gaussian, which is a stronger assumption, considering for example binary input $X$.

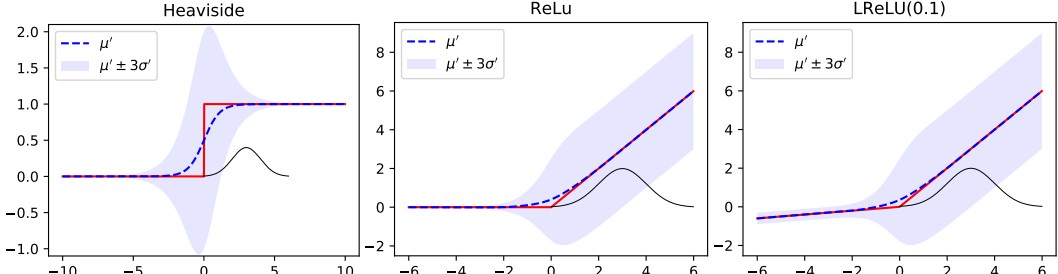

Figure 2: Propagation for the Heaviside function: $Y = [\![A \geq 0]\!]$, ReLU: $Y = \max(0, A)$ and leaky ReLU: $Y = \max(\alpha A, A)$. Red: activation function. Black: an exemplary input distribution with mean $\mu = 3$, variance $\sigma^2 = 1$ shown on the support $\mu \pm 3\sigma$. Dashed blue: the approximate mean $\mu'$ of the output versus the input mean $\mu$. The variance of the output is shown as blue shaded area $\mu' \pm 3\sigma'$.

**AP1 and AP2** The standard NN can be viewed as a further simplification of the proposed method: it makes the same factorization assumption but does not compute variances of the activations (6b) and propagates only the means. Consequently, a zero variance is assumed in propagation through non-linearities. In this case the expected values of mappings such as $\Theta(A)$ and $\mathrm{ReLU}(A)$ are just these functions evaluated at the input mean. For injected noise models we obtain smoothed versions: *e.g.*, substituting $\sigma = 0$ in the noisy Heaviside function (9) recovers the standard sigmoid function. We thus can view standard NNs as making a simpler form of factorized inference in the same Bayesian NN model. We designate this simplification (in figures and experiments) by AP1 and the method using variances by AP2 ("AP" stands for approximation).

## 4 PROPAGATION IN CATEGORICAL AND MAX LAYERS

In this section we present our main technical contribution: propagation rules for $\mathrm{argmax}$, $\mathrm{softmax}$ and $\max$ mappings, which are non-linear and multivariate. Similar to how a sigmoid function is obtained as the expectation of the Heaviside function with injected noise in Example 2, we observe that $\mathrm{softmax}$ is the expectation of $\mathrm{argmax}$ with injected noise. It follows that the standard NN with $\mathrm{softmax}$ layer can be viewed as AP1 approximation of $\mathrm{argmax}$ layer with injected noise. We propose a new approximation for the $\mathrm{argmax}$ posterior probability that takes into account uncertainty (variances) of the activations and enables propagation through $\mathrm{argmax}$ and $\mathrm{softmax}$ layers. Next, we observe that the maximum of several variables (used in max-pooling) can be expressed through $\mathrm{argmax}$. This gives a new one-shot approximation of the expected maximum using $\mathrm{argmax}$ probabilities. The logarithm of softmax, important in variational Bayesian methods can be also handled as shown in § A.2. Finally, we consider the case of leaky $\mathrm{ReLU}$, which is a maximum of two correlated variables. The proposed approximations are relatively easy to compute and are continuously differentiable, which facilitates their usage in NNs.

### 4.1 ARGMAX AND SOFTMAX

The $\mathrm{softmax}$ function, most commonly used to model a categorical distribution, thus ubiquitous in classification, is defined as $p(Y{=}y|x) = e^{x_y}/\sum_k e^{x_k}$, where $y$ is the class index. We explore the following latent variable representation known in the theory of discrete choice: $p(Y{=}y|x) = \mathbb{E}[\overline{Y}_y | X{=}x]$, where $\overline{Y} \in \{0, 1\}^n$ is the indicator of the noisy argmax: $\overline{Y}_y = [\![\mathrm{argmax}_k(X_k + \Gamma_k) = y]\!]$ and $\Gamma_k$ follow the standard Gumbel distribution. Standard NNs implement the AP1 approximation of this latent model: conditioned on $X = x$, the expectation over latent noises $\Gamma$ is the $\mathrm{softmax}(x)$.

For the AP2 approximation we need to compute the expectation w.r.t. both: $X$ and $\Gamma$, or, what is the same, to compute the expectation of $\mathrm{softmax}(X)$ over $X$. This task is difficult, particularly because variances of $X_i$ may differ across components. First, we derive an approximation for the expectation of $\mathrm{argmax}$ indicator without injected noise:

$$\overline{Y}_y = [\![\mathrm{argmax}_k X_k = y]\!]. \tag{10}$$

The injected noise case can be treated by simply increasing the variance of each $X_i$ by the variance of standard Gumbel distribution.

Let $X_k$, $k = 1, \ldots, n$ be independent, with mean $\mu_k$ and variance $\sigma_k^2$. We need to estimate

$$\mathbb{E}[\overline{Y}_y] = \mathbb{E}_X[\![X_y - X_k \geq 0 \; \forall k \neq y]\!]. \tag{11}$$

The vector $U$ with components $U_k = X_y - X_k$ for $k \neq y$ is from $\mathbb{R}^{n-1}$ with component means $\tilde{\mu}_k = \mu_y - \mu_k$ and component variances $\tilde{\sigma}_k^2 = \sigma_y^2 + \sigma_k^2$. Notice that the components of $U$ are not independent. More precisely, the covariance matrix has $\tilde{\sigma}_k^2$ on diagonal and all off-diagonal elements equal $\sigma_k^2$.

We approximate the distribution of $U$ by the $(n-1)$-variate logistic distribution defined by Malik & Abraham (1973). This choice is motivated by the following facts: its cdf $\mathcal{S}_{n-1}(u) = \frac{1}{1 + \sum_k e^{-u_k}}$ is tractable and is seen to be equivalent to the softmax function; its covariance matrix is $(I + 1)\sigma_{\mathcal{S}}^2/2$, where I is the identity matrix, *i.e.* it has similar structure to that of $U$. The approximation is made by shifting and rescaling the distribution of $U$ in order to match the means and marginal variances, *i.e.* $(U_k - \tilde{\mu}_k)\sigma_{\mathcal{S}}/\tilde{\sigma}_k$ is approximated with standard $(n-1)$-variate logistic distribution. This approximation allows to evaluate the necessary probability as

$$q(y) = \mathbb{E}[\overline{Y}_y] = \mathbb{P}(U \geq 0) = \mathbb{P}\Big(\frac{U_k - \tilde{\mu}_k}{\tilde{\sigma}_k/\sigma_{\mathcal{S}}} \geq \frac{-\tilde{\mu}_k}{\tilde{\sigma}_k/\sigma_{\mathcal{S}}} \; \forall k \neq y\Big) = \mathcal{S}_{n-1}\Big(\frac{-\tilde{\mu}_k}{\tilde{\sigma}_k/\sigma_{\mathcal{S}}}\Big). \tag{12}$$

Expanding $\tilde{\mu}$, $\tilde{\sigma}^2$ and noting that $\mu_k - \mu_y = 0$ for $y = k$, we obtain the approximation

$$q(y) = \Big(\sum_k \exp\Big\{\frac{\mu_k - \mu_y}{\sqrt{(\sigma_k^2 + \sigma_y^2)/\sigma_{\mathcal{S}}^2}}\Big\}\Big)^{-1}. \tag{13}$$

Computing this approximation has linear memory complexity but requires quadratic time in the number of inputs, which may be prohibitive for some applications.

**Linear Time Approximation**   We now derive a simpler linear-time approximation used in all our experiments. The variable $X_y$ is decomposed as $X_y = X'_y + Z$ such that $X'_y \sim \mathcal{N}(\mu_k, \sigma_a^2)$, $Z \sim \mathcal{N}(0, \sigma_y^2 - \sigma_a^2)$, where $\sigma_a$ is chosen as $\sigma_a = \min_k \sigma_k$ so that the decomposition is valid for all $k$. The variables $U_k$ are introduced as $U_k = X'_y - X_k$. The estimation of $\mathbb{E}[\overline{Y}_y]$ expresses as $\mathbb{P}(U + Z \geq 0) = \mathbb{E}_Z[\mathbb{P}(U \geq -Z|Z)]$. The probability $\mathbb{P}(U \geq -Z|Z)$ is approximated, the same way as above, by fitting $U$ with $(n-1)$-variate logistic distribution,

$$\mathbb{P}(U \geq -Z|Z) = \mathcal{S}_{n-1}\Big(\frac{-Z - \tilde{\mu}_k}{\tilde{\sigma}_k/\sigma_{\mathcal{S}}}\Big). \tag{14}$$

To achieve linear complexity, this is simplified now with approximating $\tilde{\sigma}_k^2 = \sigma_k^2 + \sigma_a^2 \approx 2\sigma_a^2$ for all $k$. It remains to integrate over $Z$. Denoting $s = \sqrt{2}\sigma_a/\sigma_{\mathcal{S}}$, we have

$$\mathbb{E}\Big[\mathcal{S}_{n-1}\Big(\frac{-Z - \tilde{\mu}_k}{s}\Big)\Big] = \mathbb{E}\Big[\frac{1}{1 + \sum_{k \neq y} \exp\big(\frac{-Z - \tilde{\mu}_k}{s}\big)}\Big] = \mathbb{E}\Big[\frac{1}{1 + e^{-Z'}}\Big], \tag{15}$$

where we denoted $Z' = Z/s + \log S$ and $S = \sum_{k \neq y} \exp(-\tilde{\mu}_k/s)$. The latter expectation in (15) is that of a regular sigmoid function, which we approximate similar to (9) as

$$\mathbb{E}\Big[\frac{1}{1 + e^{-Z'}}\Big] = \mathcal{S}\Big(\mathbb{E}[Z']/\sqrt{\text{Var}[Z']/\sigma_{\mathcal{S}}^2 + 1}\Big) = \frac{1}{1 + S^{s/s_y}}, \tag{16}$$

where $s_y = \sqrt{(\sigma_y^2 + \sigma_a^2)}/\sigma_{\mathcal{S}}$ and we used that $\mathbb{E}[Z'] = -\log S$ and $\text{Var}[Z']/\sigma_{\mathcal{S}}^2 + 1 = (\sigma_y^2 - \sigma_a^2)/s^2/\sigma_{\mathcal{S}}^2 + 1 = ((\sigma_y^2 - \sigma_a^2)/\sigma_{\mathcal{S}}^2 + s^2)/s^2 = ((\sigma_y^2 + \sigma_a^2)/\sigma_{\mathcal{S}}^2)/s^2 = s_y^2/s^2$. Expanding S in (16), as it depends on the label $y$, and rearranging we obtain the approximation:

$$q(y) = \frac{1}{1 + S^{s_y/s}} = \frac{1}{1 + \big(\sum_{k \neq y} e^{\mu_k/s} e^{-\mu_y/s}\big)^{s/r}} = \frac{e^{\mu_y/s_y}}{e^{\mu_y/s_y} + \big(\sum_{k \neq y} e^{\mu_k/s}\big)^{s/s_y}}. \tag{17}$$

This approximation is similar to softmax but reweighs the summands differently if $\sigma_y$ differs from $\sigma_a$. Clearly, it can be computed in linear time. In case when all input variances are equal, the approximation is equivalent to (13). In case when input variances are that of standard Gumbel distribution, the approximation recovers back the standard softmax of $\mu_k$.

## 4.2 Maximum of Several Variables

Let $X_k$, $k = 1, \ldots, n$ be independent, with mean $\mu_k$ and variance $\sigma_k^2$. The moments of the maximum $Y = \max_k X_k$, assuming the distributions of $X_k$ are known, can be computed from the cdf of $Y$ given by $F_Y(y) = \mathbb{P}(X_k \leq y \ \forall k) = \prod_k F_{X_k}(y)$, by numerical integration of this cdf (Ross, 2010, sec. 3.2).

We seek a simpler approximation. One option is to compose the maximum of $n > 2$ variables hierarchically using maximum of two variables (discussed below) assuming normality and independence of intermediate results. We propose a new non-trivial one-shot approximations for the mean and variance provided that the $\texttt{argmax}$ probabilities $q_k = \mathbb{P}(X_k \geq X_j \ \forall j)$ are already estimated. The derivation of these approximations and proofs of their accuracy are given in § A.1.

**Proposition 1.** Assuming $X_k$ are logistic with statistics $(\mu_k, \sigma_k^2)$, the mean of $Y = \max_k X_k$ is upper bounded by

$$\mu' \approx \sum_k q_k \hat{\mu}_k, \quad \text{where } \hat{\mu}_k = \mu_k + \frac{\sigma_k}{q_k \sigma_S} H(q_k), \tag{18}$$

where $H(q_k)$ is the entropy of the Bernoulli distribution with probability $q_k$. Notice that the entropy is non-negative, and thus $\mu'$ increases when the $\texttt{argmax}$ is ambiguous, as expected in the extreme value theory. The variance of $Y$ can be approximated as

$$\sigma'^2 \approx \sum_k \sigma_k^2 \mathcal{S}(a + b\mathcal{S}^{-1}(q_k)) + \sum_k q_k(\hat{\mu}_k - \mu')^2, \tag{19}$$

where $a = -1.33751$ and $b = 0.886763$ are coefficients originating from a Taylor expansion.

## 4.3 Maximum of Two variables and Leaky ReLU

The function $\max(X_1, X_2)$ allows to model popular leaky ReLU and maxOut layers. Although the expressions for the moments are known and have been used in the literature, *e.g.*, (Hernández-Lobato & Adams, 2015; Gast & Roth, 2018), we propose approximations that are more practical for end-to-end learning: cheap to compute and having asymptotically correct output to input variance ratio for small noises.

The exact expressions for the moments for the maximum of two Gaussian random variables $X_1, X_2$ are as follows (Nadarajah & Kotz, 2008). Denoting $s = (\sigma_1^2 + \sigma_2^2 - 2\,\mathrm{Cov}[X_1, X_2])^{\frac{1}{2}}$ and $a = (\mu_1 - \mu_2)/s$, the mean and variance of $\max(X_1, X_2)$ can be expressed as:

$$\mu' = \mu_1 \Phi(a) + \mu_2 \Phi(-a) + s\phi(a), \tag{20a}$$

$$\sigma'^2 = (\sigma_1^2 + \mu_1^2)\Phi(a) + (\sigma_2^2 + \mu_2^2)\Phi(-a) + (\mu_1 + \mu_2)s\phi(a) - \mu'^2, \tag{20b}$$

where $\phi$ and $\Phi$ are the pdf and the cdf of the standard normal distribution, resp. As $\Phi$ has to be numerically approximated with other functions, this has high computational cost and poor relative accuracy for large $|a|$. The difference of such functions occurring in (20b) may result in a negative output variance, the approximation becomes inaccurate for small noises. For the mean, we can substitute $\Phi(a)$ with an approximation such as logistic cdf $\mathcal{S}(a/\sigma_S)$. To approximate the variance, we express it as

$$\sigma'^2 = \sigma_1^2 \Phi(a) + \sigma_2^2 \Phi(-a) + s^2(a^2\Phi(a) + a\phi(a) - (a\Phi(a) + \phi(a))^2). \tag{21}$$

We observe that the function of one variable $a^2\Phi(a) + a\phi(a) - (a\Phi(a) + \phi(a))^2$ is always negative, quickly vanishes with increasing $|a|$ and is above $-0.16$. By neglecting it, we obtain a rather tight upper bound $\sigma'^2 \leq \sigma_1^2 \Phi(a) + \sigma_2^2(1 - \Phi(a))$, *i.e.*, in the form of two non-negative summands.

In case of $\texttt{LReLU}$ defined as $Y = \max(\alpha X, X)$, the variance can be approximated more accurately. Assume that $\alpha < 1$, let $X_2 = \alpha X_1$ and denote $\mu = \mu_1$ and $\sigma^2 = \sigma_1^2$. Substituting, we obtain $\mu_2 = \alpha\mu$, $\sigma_2^2 = \alpha^2\sigma^2$; $\mathrm{Cov}[X_1, X_2] = \mathrm{Cov}[X_1, \alpha X_1] = \alpha\sigma^2$; $s = \sigma(1-\alpha)$ and $a = (\mu_1 - \mu_2)/s = \mu(1-\alpha)/s = \mu/\sigma$. The mean $\mu'$ expresses as

$$\mu' = \mu(\alpha + (1-\alpha)\Phi(a)) + \sigma(1-\alpha)\phi(a). \tag{22}$$

The variance $\sigma'^2$ expresses as

$$\sigma^2\Big(\Phi(a) + \alpha^2(1-\Phi(a)) + (1-\alpha)^2\big(a^2\Phi(a) + a\phi(a) - (a\Phi(a) + \phi(a))^2\big)\Big) \tag{23}$$

$$= \sigma^2(\alpha^2 + 2\alpha(1-\alpha)\Phi(a) + (1-\alpha)^2\mathcal{R}(a)), \tag{24}$$

where $\mathcal{R}(a) = a\phi(a) + (a^2 + 1)\Phi(a) - (a\Phi(a) + \phi(a))^2$ is a sigmoid-shaped function of one variable. In practice we approximate $\sigma'^2$ with the simpler function

$$\sigma'^2 \approx \sigma^2(\alpha^2 + (1 - \alpha^2)\mathcal{S}(a/t)), \qquad (25)$$

where $t = 0.3758$ is set by fitting the approximation. The approximation is shown in Fig. 2 with more detailed evaluation given in § B.1.

## 5 EXPERIMENTS

In the experiments we evaluate the accuracy of the proposed approximation and compare it to the standard propagation. We also test the method in the end-to-end learning and show that with a simple calibration it achieves better test likelihoods than the state-of-the-art. Full details of the implementation, training protocols, used datasets and networks are given in § C. The running time of AP2 is $2\times$ more for a forward pass and 2-3$\times$ more for a forward-backward pass than that of AP1.

### 5.1 APPROXIMATION ACCURACY

The accuracy of approximations of the individual layers is evaluated in § B and is deemed sufficient for approximately propagating uncertainty and computing derivatives. We now consider multiple layers.

We conduct two experiments: how well the proposed method approximates the real posterior of neurons, w.r.t. noise in the network input and w.r.t. dropout. The first case (illustrated in Fig. 1) is studied on the LeNet5 model of LeCun et al. (2001), a 5-layer net with max pooling detailed in § C.4, trained on MNIST dataset using standard methods. We set LReLU activations with $\alpha = 0.01$ to test the proposed approximations. We estimate the ground truth statistics $\mu^*$, $\sigma^*$ of all neurons by the Monte Carlo (MC) method: drawing 1000 samples of noise per input image and collecting sample-based statistics for each neuron. Then we apply AP1 to compute $\mu_1$ and AP2 to compute $\mu_2$ and $\sigma_2$ for each unit from the clean input and known noise variance $\sigma_0^2$. The error measure of the means $\varepsilon_\mu$ is the average $|\mu - \mu^*|$ relative to the average $\sigma^*$. The averages are taken over all units in the layer and over input images. The error of the standard deviation $\varepsilon_\sigma$ is the geometric mean of $\sigma/\sigma^*$, representing the error as a factor from the true value (*e.g.*, 1.0 is exact, 0.9 is under-estimating and 1.1 is over-estimating). Table 1 shows average errors per layer. Our main observation is that AP2 is more accurate than AP1 but both methods suffer from the factorization assumption. The variance computed by AP2 provides a good estimate and the estimated categorical distribution obtained by propagating the variance through softmax is much closer to the MC estimate.

Next, we study a widely used ALL-CNN network by Springenberg et al. (2015) trained with standard dropout on CIFAR-10. Bernoulli dropout noise with dropout rate 0.2 is applied after each activation. The accuracies of estimated statistics w.r.t. dropout noises are shown in Table 2. Here, each layer receives uncertainty propagated from preceding layers, but also new noises are mixed-in in each layer, which works in favor of the factorization assumption. The results are shown in Table 2. Observe that GT noise std $\sigma^*$ changes significantly across layers, up to 1-2 orders and AP2 gives a useful estimate. Furthermore, having estimated the average factors suggests a simple calibration.

**Calibration**   We divide the std in the last layer by the average factor $\sigma/\sigma^*$ estimated on the training set. With this method, denoted AP2 calibrated, we get significantly better test likelihoods in the end-to-end learning experiment.

### 5.2 ANALYTIC NORMALIZATION

The AP2 method can be used to approximate neuron statistics w.r.t. the input chosen at random from the training dataset as was proposed by Shekhovtsov & Flach (2018). Instead of propagating sample instances, the method takes the dataset statistics $(\mu^0, (\sigma^0)^2)$ and propagates them once through all network layers, averaging over spatial dimensions. The obtained neuron mean and variance are then used to normalize the output the same way as in batch normalization (Ioffe & Szegedy, 2015). This normalization leads to a better conditioned initialization and training and is batch-independent. We verify the efficiency of this method for a network that includes the proposed approximations for LReLU and max pooling layers in § C.5 and use it in the end-to-end learning experiment below.

| | Conv | LReLU | MaxPool | Conv | LReLU | MaxPool | FC | LReLU | FC | LReLU | FC | Softmax |
|---|---|---|---|---|---|---|---|---|---|---|---|---|
| Noisy input with noise std $\sigma_0 = 10^{-2}$ | | | | | | | | | | | | |
| $\sigma^*$ | 0.03 | 0.02 | 0.02 | 0.06 | 0.03 | 0.03 | 0.09 | 0.05 | 0.10 | 0.05 | 0.11 | |
| $\varepsilon_{\mu_1}$ | 0.02 | 0.19 | 0.37 | 0.84 | 0.43 | 0.52 | 1.20 | 1.16 | 0.62 | | 1.25 | KL 3.5e-4 |
| $\varepsilon_{\mu_2}$ | 0.02 | 0.02 | 0.13 | 0.29 | 0.13 | 0.17 | 0.37 | 0.21 | 0.36 | 0.20 | 0.39 | KL 3.3e-5 |
| $\varepsilon_{\sigma_2}$ | 1.00 | 1.05 | 1.25 | 1.06 | 1.06 | 1.12 | 1.09 | 1.10 | 1.03 | 1.04 | 0.96 | |
| Noisy input with noise std $\sigma_0 = 10^{-1}$ | | | | | | | | | | | | |
| $\sigma^*$ | 0.3 | 0.16 | 0.20 | 0.58 | 0.24 | 0.27 | 0.79 | 0.47 | 0.86 | 0.42 | 0.92 | |
| $\varepsilon_{\mu_1}$ | 0.02 | 0.24 | 0.53 | 1.46 | 0.58 | 0.70 | 1.44 | 0.85 | 1.40 | 0.79 | 1.57 | KL 0.36 |
| $\varepsilon_{\mu_2}$ | 0.02 | 0.02 | 0.21 | 0.65 | 0.21 | 0.31 | 0.61 | 0.37 | 0.67 | 0.34 | 0.72 | KL 0.05 |
| $\varepsilon_{\sigma_2}$ | 1.00 | 1.10 | 1.15 | 1.17 | 1.22 | 1.42 | 1.37 | 1.59 | 1.31 | 1.47 | 1.23 | |

Table 1: Accuracy of approximation of mean and variance statistics for each layer in a fully trained LeNet5 (MNIST) tested with noisy input. Observe the following: MC std $\sigma^*$ is growing significantly from the input to the output; both AP1 and AP2 have a significant drop of accuracy at linear (FC and Conv) layers, due to factorized approximation assumption; AP2 approximation of the standard deviation is within a factor close to one, and makes a meaningful estimate, although degrading with depth; AP2 approximation of the mean is more accurate than AP1; the KL divergence from the MC class posterior is improved with AP2.

| | C | A | C | A | C | A | C | A | C | A | C | A | C | A | C | A | C | P | Softmax |
|---|---|---|---|---|---|---|---|---|---|---|---|---|---|---|---|---|---|---|---|
| $\sigma^*$ | 0 | 0.26 | 0.31 | 0.46 | 0.86 | 0.77 | 1.1 | 0.78 | 1.7 | 0.97 | 2.2 | 1.3 | 1.5 | 0.89 | 2 | 0.74 | 16 | 2.8 | |
| $\varepsilon_{\mu_1}$ | - | 0.01 | 0.02 | 0.03 | 0.07 | 0.06 | 0.17 | 0.09 | 0.19 | 0.10 | 0.25 | 0.11 | 0.22 | 0.11 | 0.21 | 0.12 | 0.17 | 0.38 | KL 0.11 |
| $\varepsilon_{\mu_2}$ | - | 0.01 | 0.02 | 0.01 | 0.02 | 0.02 | 0.05 | 0.02 | 0.06 | 0.03 | 0.07 | 0.04 | 0.08 | 0.04 | 0.09 | 0.04 | 0.05 | 0.14 | KL 0.04 |
| $\varepsilon_{\sigma_2}$ | - | 1.00 | 1.00 | 1.02 | 0.88 | 0.89 | 0.90 | 0.95 | 0.84 | 0.87 | 0.77 | 0.77 | 0.82 | 0.85 | 0.88 | 0.92 | 0.69 | 0.45 | |

Table 2: Accuracy of approximation of mean and variance statistics for each layer in All-CNN (CIFAR-10) trained and tested with dropout. The table shows accuracies after all layers (C-convolution, A-activation, P-average pooling) and the final KL divergence. A similar effect to propagating input noise is observed: the MC std $\sigma^*$ grows with depth; a significant drop of accuracy is observed in convolutional and pooling layers, which rely on the independence assumption.

## 5.3 END-TO-END LEARNING WITH ANALYTIC DROPOUT

In this experiment we approximate the dropout analytically at training time similar to Wang & Manning (2013) but including the new approximations for LReLU and softmax layers. We compare training All-CNN network on CIFAR-10 without dropout, with standard dropout (Srivastava et al., 2014) and analytic (AP2) dropout. All three cases use exactly the same initialization, the AP2 normalization as discussed above and the same learning setup. Only the learning rate is optimized individually per method § C.3. Dropout layers with dropout rate 0.2 are applied after every activation. Fig. 3 shows the progress of the three methods. The analytic dropout is efficient as a regularizer (reduces overfitting in the validation likelihood), is non-stochastic and allows for faster learning than standard dropout. While the latter slows the training down due to increased stochasticity of the gradient, the analytic dropout smoothes the loss function and speeds the training up. This is especially visible on the training loss plot in Fig. C.3. Furthermore, analytic dropout can be applied as the test-time inference method in a network trained with any variant of dropout. Table 3 shows that AP2, calibrated as proposed above, achieves the best test likelihood, significantly improving SOA results for this network. Differently from Wang & Manning (2013), we find that when trained with standard dropout, all test methods achieve approximately the same accuracy and only differ in likelihoods.

We also attempted comparison with other approaches. Gaussian dropout (Srivastava et al., 2014) performed similarly or slightly worse than Bernoulli dropout. Variational dropout (Kingma et al., 2015) in our implementation for convolutional networks has diverged or has not improved over the no-dropout baseline (we tried correlated and uncorrelated versions with or without local reparametrization trick and with different KL divergence factors $1, 0.1, 0.01, 0.001$).

## 6 CONCLUSION

We have described uncertainty propagation method for approximate inference in probabilistic neural networks that takes into account all noises analytically. Latent variable models allow a transparent interpretation of standard propagation in NNs as the simplest approximation and facilitate the devel-

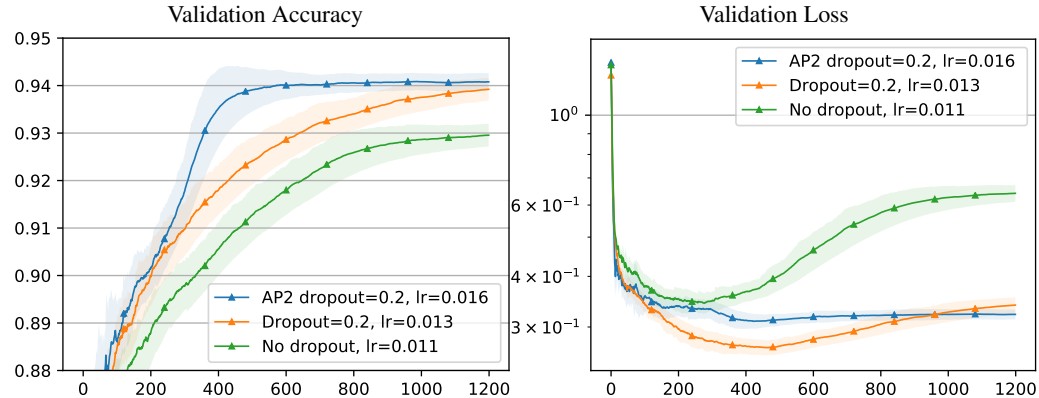

Figure 3: Comparison of analytic AP2 dropout with baselines. All methods use AP2 normalization during training. Analytic dropout converges to similar values of stochastic dropout and is faster in iterations. Both methods are efficient in preventing overfitting as seen in the right plot.

| SOTA results (Gast & Roth, 2018) | | | Standard dropout | | | Analytic dropout | | |
|---|---|---|---|---|---|---|---|---|
| Method | NLL | Acc. | Test method | NLL | Acc. | Test method | NLL | Acc. |
| Dropout MC-30 | 0.327 | 90.88 | AP1 | 0.434 | 0.938 | AP1 | 1.86 | **0.940** |
| ProbOut | 0.37 | 91.9 | AP2 | 0.311 | 0.936 | AP2 | 0.363 | **0.940** |
| | | | AP2 calibrated | 0.214 | 0.937 | AP2 calibrated | **0.194** | **0.940** |
| | | | MC-10 | 0.264 | 0.935 | MC-10 | 0.546 | 0.919 |
| | | | MC-100 | 0.217 | 0.937 | MC-100 | 0.281 | 0.925 |
| | | | MC-1000 | 0.210 | 0.937 | MC-1000 | 0.243 | 0.926 |

Table 3: Results for All-CNN on CIFAR-10 *test* set: negative log likelihood (NLL) and accuracy. *Left:* state of the art results for this network (Gast & Roth, 2018, table 3). *Middle:* All-CNN trained with standard dropout (our learning schedule and analytic normalization) evaluated using different test-time methods. Observe that "AP2 calibrated" well approximates dropout: the test likelihood is better than MC-100. *Right:* All-CNN trained with analytic dropout (same schedule and normalization). Observe that "AP2 calibrated" achieves the best likelihood and accuracy.

opment of variance propagating approximations. We proposed new such approximations allowing to handle max, argmax, softmax and log-softmax layers using latent variable models (§ 4 and § A.2).

We measured the quality of the approximation of posterior in isolated layers and complete networks. The accuracy is improved compared to standard propagation and is sufficient for several use cases such as estimating statistics over the dataset (normalization) and dropout training, where we report improved test likelihoods. We identified the factorization assumption as the weakest point of the approximation. While modeling of correlations is possible (*e.g.* Rezende & Mohamed, 2015), it is also more expensive. We showed that a calibration of a cheap method can give a significant improvement and thus is a promising direction for further research. Argmax and softmax may occur not only as the final layer but also inside the network, in models such as capsules (Sabour et al., 2017) or multiple hypothesis (Ilg et al., 2018), *etc*. Further applications of the developed technique may include generative and semi-supervised learning and Bayesian model estimation.

## ACKNOWLEDGMENTS

A.S. was supported by the project "International Mobility of Researchers MSCA-IF II at CTU in Prague" (CZ.02.2.69/0.0/0.0/18_070/0010457). B.F. gratefully acknowledges support by the Czech OP VVV project "Research Center for Informatics" (CZ.02.1.01/0.0/0.0/16_019/0000765).

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

# Appendix

## A   MAXIMUM AND SMOOTH MAXIMUM

### A.1   MAXIMUM OF SEVERAL VARIABLES

**Approximation of the Mean**   For each $k$ let $A_k \subset \Omega$ denote the event that $X_k > X_j \; \forall j$, *i.e.* that $X_k$ is the maximum of all variables. Let $q_k = \mathbb{P}(A_k)$ be given. Note that events $\{A_k\}_k$ partition the probability space. The expected value of the maximum $Y = \max_k X_k$ can be written as the following total expectation:

$$\mu' = \mathbb{E}\big[Y\big] = \sum_k \mathbb{P}(A_k)\mathbb{E}[Y \mid A_k] = \sum_k q_k \mathbb{E}[X_k \mid A_k]. \tag{26}$$

In order to compute each conditional expectation, we approximate the conditional density $p(X_k = x_k \mid A_k)$, which is the marginal of the joint conditional density $p(X = x \mid A_k)$, *i.e.* the distribution of $X$ restricted to the part of the probability space $A_k$ as illustrated in Fig. A.1. The approximation is a simpler conditional density $p(X_k = x_k \mid \hat{A}_k)$ where $\hat{A}_k$ is chosen in the form $\hat{A}_k = [\![X_k \geq m_k]\!]$ and the threshold $m_k$ is chosen to satisfy the proportionality:

$$\mathbb{P}(\hat{A}_k) = \mathbb{P}(A_k) = q_k, \tag{27}$$

which implies $m_k = F_{X_k}^{-1}(q_k)$. This can be also seen as the approximation of the conditional probability $\mathbb{P}(A_k \mid X_k = r) = \prod_{j \neq k} F_{X_j}(r)$, as a function of $r$, with the indicator $[\![m_k \leq r]\!]$, *i.e.* the smooth step function given by the product of sigmoid-like functions $F_{X_k}(r)$ with a sharp step function.

Assuming $X_k$ is logistic, we find $m_k = \mu_k + \sigma_k/\sigma_\mathcal{S} \log(\frac{1-q_k}{q_k})$. Then the conditional expectation $\hat{\mu}_k = E[X_k \mid \hat{A}_k]$ is computed as

$$\hat{\mu}_k = \frac{1}{q_k} \int_{m_k}^{\infty} x p(X_k{=}x)dx = \frac{1}{q_k} \int_{\log(\frac{1-q_k}{q_k})}^{\infty} (\mu_k + a\frac{\sigma_k}{\sigma_\mathcal{S}})p_S(a)da = \mu_k + \frac{1}{q_k}\frac{\sigma_k}{\sigma_\mathcal{S}}H(q_k), \tag{28}$$

where $p_S$ is the density of the standard Logistic distribution, $a = \frac{x-\mu_k}{\sigma_k/\sigma_\mathcal{S}}$ is the changed variable under the integral and $H(q_k) = -q_k \log(q_k) - (1 - q_k)\log(1 - q_k)$ is the entropy of a Bernoulli variable with probability $q_k$. This results in the following interesting formula for the mean:

$$\mu' \approx \sum_k q_k \mu_k + \sum_k \frac{\sigma_k}{\sigma_\mathcal{S}}H(q_k). \tag{29}$$

Assuming $X_k$ is normal, we obtain the approximation

$$\mu' \approx \sum_k q_k \mu_k + \sum_k \sigma_k \phi(\Phi^{-1}(q_k)). \tag{30}$$

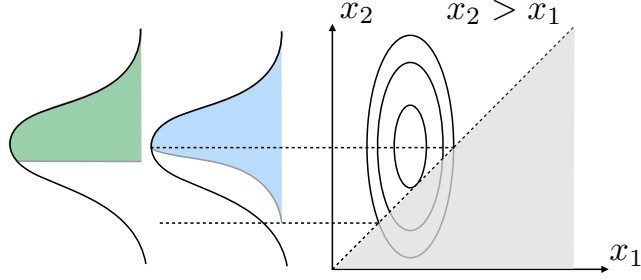

Figure A.1:   The joint conditional density $p(X_1 = x_1, X_2 = x_2 \mid X_2 > X_1)$, its marginal density $p(X_2 = x_2 \mid X_2 > X_1)$ and the approximation $p(X_2 = x_2 \mid X_2 > m_2)$, all up to the same normalization factor $\mathbb{P}(X_2 > X_1)$.

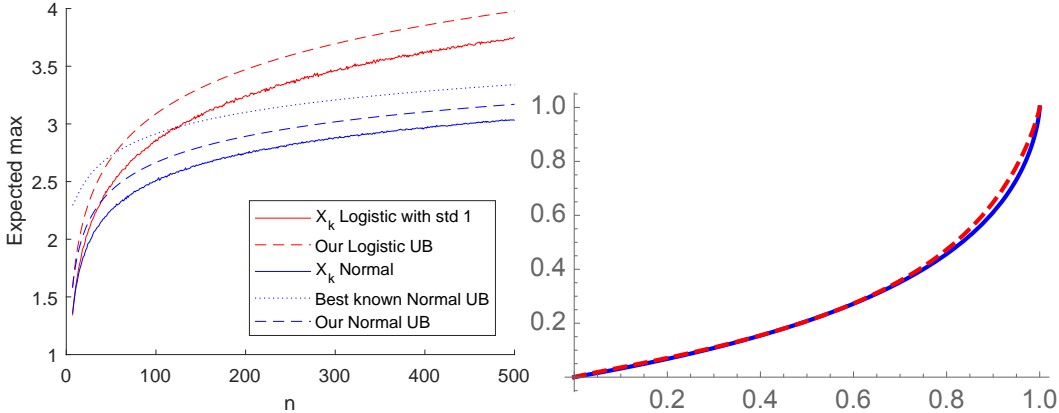

Figure A.2: *Left*: expectation of $Y = \max_k X_k$ for $X_k$ iid logistic or normal, our estimates (dashed) versus sampling-based ground truth (solid) and the best known closed form upper bound for the normal iid case (DasGupta et al., 2014, Theorem 4.1) (dotted). *Right*: the variance scaling function $f(q)$ (35) (solid) and its approximation (36) (dashed).

**Lemma A.1.** The approximation $\hat{\mu}_k$ is an upper bound on $E[X_k|A_k]$.

*Proof.* We need to show that $E[X_k|A_k] \leq E[X_k|\hat{A}_k]$. Since $\mathbb{P}(A_k) = \mathbb{P}(\hat{A}_k)$, it is sufficient to prove that

$$\int_{A_k} X_k(\omega)\mathrm{d}\mathbb{P}(\omega) \leq \int_{\hat{A}_k} X_k(\omega)\mathrm{d}\mathbb{P}(\omega). \tag{31}$$

Let us subtract the integral over the common part $A_k \cap \hat{A}_k$. It remains to show

$$\int_{A_k \setminus \hat{A}_k} X_k(\omega)\mathrm{d}\mathbb{P}(\omega) \leq \int_{\hat{A}_k \setminus A_k} X_k(\omega)\mathrm{d}\mathbb{P}(\omega). \tag{32}$$

In the RHS integral we have $X_k(\omega) \geq m_k$ since $\omega \in \hat{A}_k = \{\omega \mid X_k(\omega) \geq m_k\}$. In the LHS integral we have $X_k(\omega) < m_k$ since $\omega \notin \hat{A}_k$. Notice also that $\mathbb{P}(A_k \setminus \hat{A}_k) = \mathbb{P}(\hat{A}_k \setminus A_k)$. The inequality (32) follows.

**Corollary A.1.** The approximations of the expected maximum (29), (30) are upper bounds in the respective cases when $X_k$ are logistic, resp., normal.

Consider the case that $X_k$ are i.i.d., all logistic or normal with $\mu_k = 0$ and $\sigma_k = 1$. We then have $q_k = \frac{1}{n}$. For the logistic case $\mu' \approx nH(\frac{1}{n})$, which is asymptotically $\log(n) + 1 - \frac{1}{2n} + O(1/n^2)$. For the normal case $\mu' \approx n\phi(\Phi^{-1}(\frac{1}{n}))$. Fig. A.2 shows comparisons of these estimates with the sampling-based ground truth.

**Approximation of the Variance**   For the variance we write

$$\sigma'^2 = \mathbb{E}(Y - \mu')^2 = \sum_k q_k \mathbb{E}((X_k - \mu')^2 \mid A_k) \approx \sum_k q_k \mathbb{E}((X_k - \mu')^2 \mid \hat{A}_k), \tag{33}$$

where the approximation is due to $\hat{A}_k$, and further rewrite the expression as

$$= \sum_k q_k \mathbb{E}(X_k^2 - 2X_k\mu' + \mu'^2 \mid \hat{A}_k) \tag{34a}$$

$$= \sum_k q_k \left( \mathbb{E}(X_k^2 - \hat{\mu}_k^2 \mid \hat{A}_k) + (\hat{\mu}_k - \mu')^2 \right) \tag{34b}$$

$$= \sum_k q_k (\hat{\sigma}_k^2 + (\hat{\mu}_k - \mu')^2) \tag{34c}$$

where $\hat{\sigma}_k^2 = \text{Var}[X_k \mid \hat{A}_k]$. For $X_k$ with logistic density $p(x)$ the variance integral $\hat{\sigma}_k^2 = \int_{m_k}^{\infty}(x - \hat{\mu}')^2 p(x)\mathrm{d}x$ expresses as[2]:

$$\hat{\sigma}_k^2 = \frac{1}{q_k}\frac{\sigma_k^2}{\sigma_{\mathcal{S}}^2}\Big(-\frac{\log^2(1-q_k)}{q_k} - 2\,\text{Li}_2\big(\frac{q_k}{q_k - 1}\big)\Big) =: \frac{1}{q_k}\sigma_k^2 f(q_k), \tag{35}$$

where $\text{Li}_2$ is dilogarithm. The function $f$ can be well approximated on $[0, 1]$ with

$$\tilde{f}(q) = \mathcal{S}(a + b\mathcal{S}^{-1}(q)), \tag{36}$$

where $a = -1.33751$ and $b = 0.886763$ are obtained from the first order Tailor expansion of $\mathcal{S}^{-1}(f(\mathcal{S}(t)))$ at $t = 0$. This approximation is shown in Fig. A.2 and is in fact an upper bound on $f$. We thus obtained a rather simple approximation for the variance[3]

$$\sigma'^2 \approx \sum_k \sigma_k^2 \mathcal{S}(a + b\mathcal{S}^{-1}(q_k)) + \sum_k q_k(\hat{\mu}_k - \mu')^2. \tag{37}$$

## A.2   SMOOTH MAXIMUM – LOGSUMEXP

In variational Bayesian learning it is necessary to compute the expectation of $\log p(y|x, \theta)$ w.r.t. to random parameters $\theta$. The expectation of the logarithm rather that the logarithm of expectation originates from the variational lower bound on the marginal likelihood obtained with Jensen's inequality. In this section we extend our approximations to also handle $\log p(y|x, \theta)$ for classification problems. The same propagation rules apply up to the difference that the last layer is log of softmax rather than softmax, *i.e.*

$$\mathbb{E}[\log \text{softmax}(X)] = \mathbb{E}[X - \log \sum_k \exp X_k] = \mathbb{E}[X] - \mathbb{E}[\log \sum_k \exp X_k]. \tag{38}$$

It remains therefore to handle the log-sum-exp operation, also known as the smooth maximum.

**Proposition A.1.** The LogSumExp operation has the following latent variable representation:

$$\log \sum_k \exp(x_k) = \mathbb{E}\big[\max_k(x_k + \Gamma_k)\big], \tag{39}$$

where $\Gamma_k$ are independent Gumbel random variables such that $\mathbb{E}[\Gamma_k] = 0$, *i.e.*, $\Gamma_k \sim \text{Gumbel}(-\gamma, 1)$, where $\gamma$ is the Euler-Mascheroni constant.

*Proof.* Let $Y = \max_k(x_k + \Gamma_k)$. Recall that the cdf of $\Gamma_k$ is $F_{\text{Gumbel}(-\gamma,1)}(x) = e^{-e^{-(x+\gamma)}}$. We can write the cdf of $Y$ as

$$F_Y(y) = \mathbb{P}(x_k + \Gamma_k \le y) = \prod_k \mathbb{P}(\Gamma_k \le y - x_k) = \prod_k e^{-e^{-(y-x_k+\gamma)}} \tag{40a}$$

$$= \exp\Big(-\sum_k e^{x_k}e^{-(y+\gamma)}\Big) = \exp\Big(-e^S e^{-(y+\gamma)}\Big) = e^{-e^{-(y+\gamma-S)}} = F_{\text{Gumbel}(S-\gamma,1)}(y), \tag{40b}$$

where $S = \log\sum_k e^{x_k}$. It follows that the mean value of $Y$ is $S - \gamma + \gamma = S$.

We therefore propose to approximate the expectation of $\log\text{softmax}(X)$ by increasing the variances of all inputs $X_k$ by $\sigma_{\mathcal{S}}^2/2$ and applying the approximation for the maximum (A.1). Summarizing, we obtain the following.

**Proposition A.2.** Let $X_i$ have statistics $(\mu_i, \sigma_i^2)$. Then using expressions (39) and (29),

$$\mathbb{E}[\log\text{softmax}(X)]_j \approx \mu_j - \sum_k q_k\mu_k + \sum_k H(q_k)\sqrt{\sigma_k^2/\sigma_{\mathcal{S}}^2 + \frac{1}{2}}, \tag{41}$$

where $q_k$ are the expected softmax values § 4.1.

---

[2]Computed with the help of Mathematica.
[3]Not an upper bound due to (33).

**Two classes** For $p(y = 1|x) = 1/(1 + e^{-x})$, we have $\log p(y = 1|x) = -\log(1 + e^{-x})$. The analogue of Proposition A.1 is the latent variable expression

$$\log(1 + e^x) = \mathbb{E}[\max(0, x + Z)], \tag{42}$$

where $Z$ is a standard Logistic r.v. Therefore, to approximate $\mathbb{E}[\log(1 + e^X)]$, we can increase the variance of $X$ by the variance of standard logistic distribution $\sigma_S^2$ and apply the existing approximation for ReLU § 4.3.

## B ACCURACY OF INDIVIDUAL BLOCKS

### B.1 LEAKY RELU

We evaluate the simplified approximation of Leaky ReLU (25), which does not use the normal cdf function. Since LReLU is 1-homogenous, it is clear that scaling the input will scale the output proportionally. We therefore fix the input variance to 1 and plot the results as the function of the input mean $\mu$. Fig. B.1 shows that the approximation of the mean and variance as well as the approximation of the output distribution defined by these values are all reasonable. Fig. B.2 shows the implied approximation of derivatives. The baseline for the derivatives is the MC estimate with pathwise derivative method (PD) (Glasserman, 2004), also known as the reparametrization trick. This is also the method for the ground truth (with $10^5$ samples). Despite the approximation of the variance and its gradients are somewhat deviating from the GT model that assumes the perfect normal distribution on the input, the overall behavior of the function is similar to the desired one and it makes a cheap computational element for NNs.

### B.2 SOFTMAX

To evaluate the proposed approximation for softmax we perform the following experiment. We consider $n = 10$ inputs $X_1, \ldots, X_n$ to be independent with $X_k \sim \mathcal{N}(\mu_k, \sigma_k^2)$. The means $\mu_i$ are generated uniformly in the interval $[0, U]$. Then we sample $\sigma_k$ such that $\log \sigma_k$ is uniform in the interval $[-5, 0]$. We then estimate the ground truth output class distribution $q(y) = (\text{softmax}(X))_y$ by MC sampling using $10^5$ samples of $X$ and evaluate the KL divergence from this GT estimate to the approximations. We test both: quadratic time approximation (13) as well as linear time approximation (17). The evaluation is repeated with scaled variances, $\sigma$ such that $\max_k(\sigma_k)$ ranges from $10^{-3}U$ to $U$, *i.e.*, covers the practically relevant interval. The experiment is repeated 1000 trials in which different samples of $\mu$ and $\sigma$ are evaluated. As a baseline we take the AP1 approximation and MC sampling using fewer samples (10 and 100). This evaluation is shown in Fig. B.3 (a,d).

We further check how well the Jacobian is approximated. The Jacobian has two parts: $J^\mu_{y,k} = \partial q(y)/\partial \mu_k$ and $J^\sigma_{y,k} = \partial q(y)/\partial \sigma_k$. For each part we compute the average cosine similarity of the

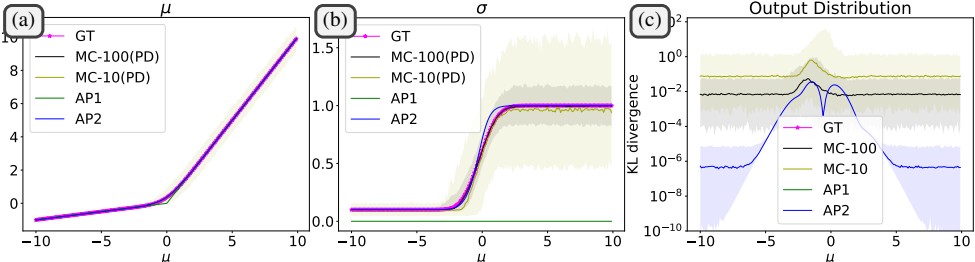

Figure B.1: Evaluation of accuracy of propagating uncertainty through Leaky ReLU(0.1). (a), (b) Approximation of the output mean and standard deviation, respectively. (c) KL divergence from the normal distribution with the ground truth mean and variance estimates. The shaded area shows where 50% of the trials fall. A trial computes the MC ground truth ($10^5$ samples) and MC baseline estimates.

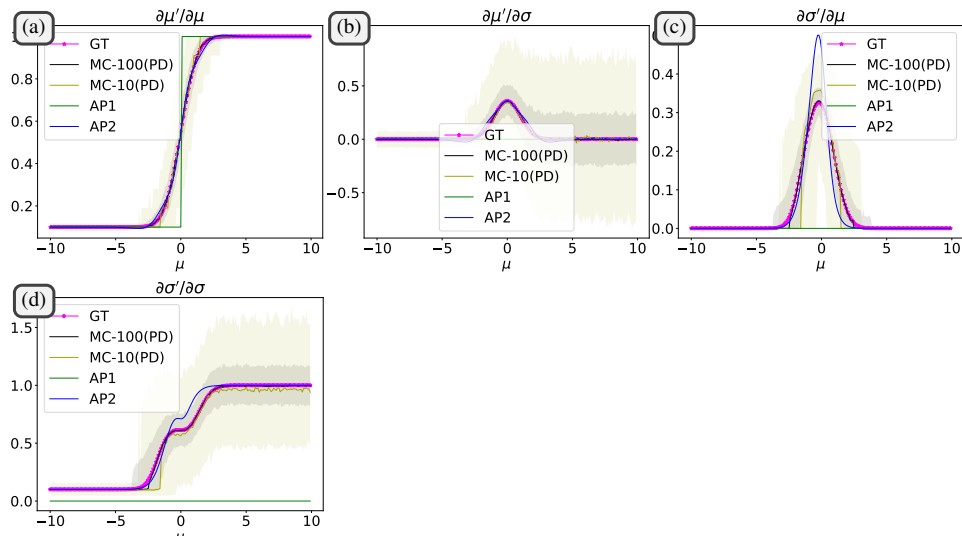

Figure B.2: Approximation of Jacobian components for LReLU(0.1). The baseline MC method uses pathwise derivatives (PD) and has a significant variance even with 100 samples. The shaded area shows the interval of $50\%$ of the trials.

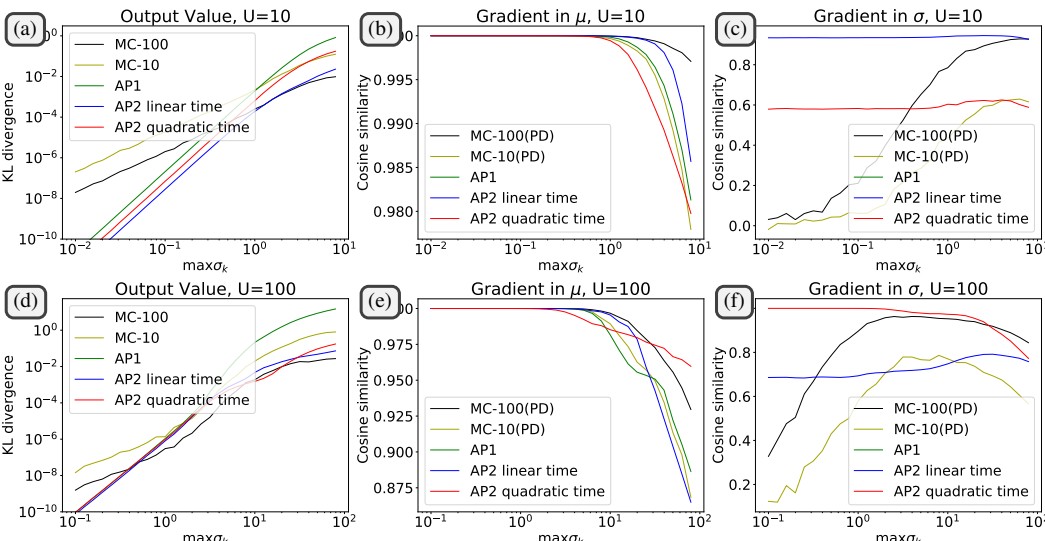

Figure B.3: Evaluation of accuracy of propagating uncertainty through softmax. Top and bottom display the cases when $\mu$ are chosen at random in intervals $[0, 10]$ and $[0, 100]$, respectively. (a,d) Approximation quality of the expected value, measured by KL divergence from GT MC estimate. Each line shows the median of the respective method over 100 trials. Analytic approximations are on par with Monte Carlo estimates and more accurate for small input noise. (b,c,e,f) Cosine similarity of gradients to the GT estimate. Each line shows the median over 1000 trials. MC estimates use pathwise derivative (PD) method. While the cosine similarity is high for all methods in gradient w.r.t. $\mu$ (notice the axis limits), analytic approximations are more accurate than MC estimates in the gradient w.r.t. $\sigma$.

gradients:

$$\frac{1}{n}\sum_y \frac{\langle J_y^{GT}, J_y \rangle}{\|J_y^{GT}\|\|J_y\|}, \tag{43}$$

where $J_y$ denotes the gradient of output $y$ in the part of the inputs ($\mu$ or $\sigma$). The baseline AP1 obviously cannot estimate the gradient in $\sigma$. This evaluation is shown in Fig. B.3 (middle, right).

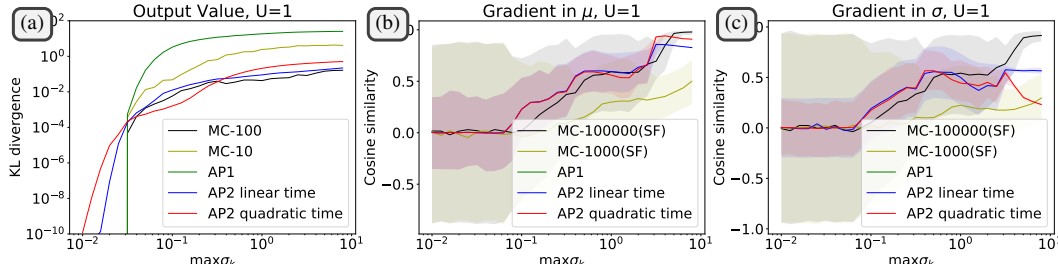

Figure B.4: Evaluation of accuracy of propagating uncertainty through Argmax. (a) Approximation quality of the expected value, measured by KL divergence from GT MC estimate. Each line shows the median of the respective method over 100 trials. (b,c) Cosine similarity of gradient orientations to the GT estimate. Each line shows the median over 1000 trials of estimation and the shaded area shows intervals where 50% of the trials fall. Each trial consists of estimating the GT using $10^7$ samples and repeating each of the MC estimates. The MC methods use score function (SF) estimator. The intervals around analytic estimates are due to the variance of the ground truth. Towards smaller input variance the ground truth estimate of the gradient degrades to completely random and the scalar product with it approaches zero on average. It does not imply that our analytic estimates are poor for small input variance.

## B.3 ARGMAX

This experiment is similar to Softmax but with several differences. Unlike in softmax, the range of $\mu$ is not important (as there is no latent logistic noise with fixed variance added to the inputs). We therefore can fix $U = 1$ because scaling both $\mu$ and $\sigma$ is guaranteed to give the same output distribution. Approximating the value of expected argmax indicator is shown in Fig. B.4(a). In the baseline methods we include AP1, which computes $y^* = \text{argmax}_k \mu_k$ and assigns the output probability $q(y^*) = 1$ and $q(y) = 10^{-20}$ for $y \neq y^*$. It is seen that the proposed approximations accurately model the expected value. Computing the gradient with MC methods is more difficult in this case, since the pathwise derivative cannot be applied (because argmax indicator is not differentiable). We therefore used the score function (SF) estimator (Fu, 2006), also known as REINFORCE method. This method requires much more samples. In fact we had problems to get a reliable ground truth even with as many as $10^7$ samples. In Fig. B.4(b,c) we illustrate the gradient estimation for a single random instance of $\mu, \sigma$. These plots show that baseline MC estimates have very high variance with $10^3$ and $10^5$ samples used. The accuracy of the gradients with analytic method for small variances remains largely unmeasured because the GT estimate also degrades quickly and becomes close to random for small input noises. A more accurate GT could be possibly computed by variance reduction techniques, in particular using our analytic estimates as (biased) baselines.

## C MULTILAYER EXPERIMENTS DETAILS

In this section we give all details necessary to ensure reproducibility of results.

### C.1 IMPLEMENTATION DETAILS

We implemented our inference and learning in the pytorch[4] framework. The source code will be publicly available. The implementation is modular: with each of the standard layers we can do 3 kinds of propagation: *AP1*: standard propagation in deterministic layers and taking the mean in stochastic layers (*e.g.*, in dropout we need to multiply by the Bernoulli probability), *AP2*: proposed propagation rules with variances and *sample*: by drawing samples of any encountered stochasticity (such as sampling from Bernoulli distribution in dropout). The last method is also essential for computing Monte Carlo (MC) estimates of the statistics we want to approximate. When the training method is *sample*, the test method is assumed to be AP1, which matches the standard practice of dropout training.

---

[4]http://pytorch.org

In the implementation of AP2 propagation the input and the output of each layer is a pair of mean and variance. At present we use only higher-level pytorch functions to implement AP2 propagation. For example, AP2 propagation for convolutional layer is implemented simply as

```
y.mean = F.conv2d(x.mean, w) + b
y.var = F.conv2d(x.var, w*w)
```

For numerical stability, it was essential that logsumexp is implemented by subtracting the maximum value before exponentiation

```
m, _ = x.max()
m = m.detach() # does not influence gradient
y = m + torch.log(torch.sum(torch.exp(x − m)))
```

The feed-forward propagation with AP2 is about 3 times slower than AP1 or sample. The relative times of a forward-backward computation in our higher-level implementation are as follows:

```
standard training       1
BN                      1.5
inference=AP2           3
inference=AP2−norm=AP2  6
```

Please note that these times hold for unoptimized implementations. In particular, the computational cost of the AP2 normalization, which propagates single pixel statistics, should be more efficient in comparison to propagating a batch of input images.

## C.2  DATASETS

We used MNIST[5] and CIFAR10[6] datasets. Both datasets provide a split into training and test sets. From the training set we split 10 percent (at random) to create a validation set. The validation set is meant for model selection and monitoring the validation loss and accuracy during learning. The test sets were currently used only in the stability tests.

## C.3  TRAINING

For the optimization we used batch size 32, SGD optimizer with Nesterov Momentum 0.9 (pytorch default) and the learning rate $lr \cdot \gamma^k$, where $k$ is the epoch number, $lr$ is the initial learning rate, $\gamma$ is the decrease factor. In all reported results for CIFAR we used $\gamma$ such that $\gamma^{600} = 0.1$ and 1200 epochs. This is done in order to make sure we are not so much constrained by the performance of the optimization and all methods are given sufficient iterations to converge. The initial learning rate was selected by an automatic numerical search optimizing the training loss in 5 epochs. This is performed individually per training case to take care for the differences introduced by different initializations and training methods.

When not said otherwise, parameters of linear and convolutional layers were initialized using pytorch defaults, *i.e.*, uniformly distributed in $[−1/\sqrt{c}, 1/\sqrt{c}]$, where c is the number of inputs per one output.

Standard minor data augmentation was applied to the training and validation sets in CIFAR-10, consisting in random translations $\pm 2$ pixels (with zero padding) and horizontal flipping.

When we train with normalization, it is introduced after each convolutional and fully connected layer.

## C.4  NETWORK SPECIFICATIONS

The LeNet5 architecture LeCun et al. (2001) is:

```
Conv2d(1, 6, ks=5, st=2), Activation
MaxPooling
```

---

[5] http://yann.lecun.com/exdb/mnist/
[6] https://www.cs.toronto.edu/~kriz/cifar.html

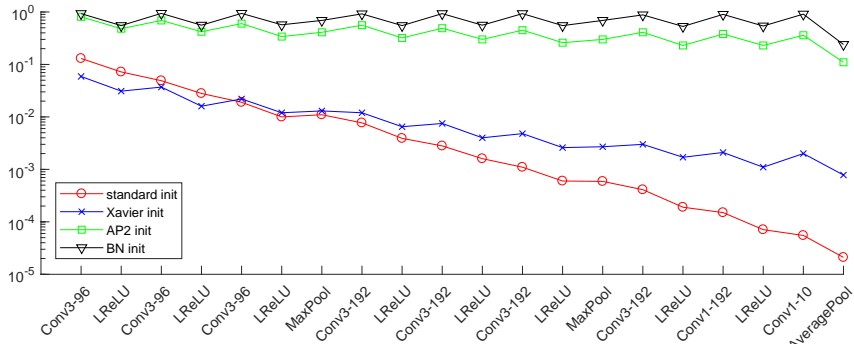

Figure C.1: Standard deviation of neurons in network layers after different initializations. The shown values are averages over all units in each layer (spatial and channel dimensions). With standard random initialization the variances quickly decrease and the network output for the whole dataset collapses nearly to a single point, complicating the training. Xavier init does not fully resolve the problem. Analytic normalization provides standard deviation within a small factor of 1 in all layers, comparable to BN. The zig-zagging effect is observed because the normalization is performed after linear layers only.

```
Conv2d(6, 16, ks=5, st=2), Activation
MaxPooling
FC(4*4*16, 120), Activation
FC(120, 84), Activation
FC(84, 10), Activation
LogSoftmax
```

Convolutional layer parameters list input channels, output channels, kernel size and stride.

The *All-CNN* network Springenberg et al. (2015) has the following structure of convolutional layers:

```
ksize = [3,   3,   3,   3,    3,    3,    3,    1,    1 ]
stride= [1,   1,   2,   1,    1,    2,    1,    1,    1 ]
depth = [96,  96,  96,  192,  192,  192,  192,  192,  10]
```

each but the last one ending with activation (we used LReLU). The final layers of the network are

```
AdaptiveAvgPool2d, LogSoftmax
```

*ConvPool-CNN-C* model replaces stride-2 convolutions by stride-1 convolutions of the same shape followed by 2x2 max pooling with stride 2.

## C.5   Auxiliary Results on Normalization

We test the analytic normalization method (Shekhovtsov & Flach, 2018) in a network with max pooling and Leaky ReLU layers. We consider the "ConvPool-CNN-C" model of Springenberg et al. (2015) on CIFAR-10 dataset. It's structure is shown on the x-axis of Fig. C.1. We first apply different initialization methods and compute variances in each layer over the training dataset. Fig. C.1 shows that standard initialization with weights distributed uniformly in $[-1/\sqrt{n_{in}}, 1/\sqrt{n_{in}}]$, where $n_{in}$ is the number of inputs per single output of a linear mapping results in the whole dataset concentrated around one output point with standard deviation $10^{-5}$. Initialization of Glorot & Bengio (2010), using statistical arguments, improves this behavior. For the analytic approximation, we take statistics of the dataset itself $(\mu_0, \sigma_0)$ and propagate them through the network, ignoring spatial dimensions of the layers. When normalized by this estimates, the real dataset statistics have variances close to one and means close to zero, *i.e.* the normalization is efficient. For comparison, we also show normalization by the batch statistics with a batch of size 32. Fig. C.2 further demonstrates that the initialization is crucial for efficient learning, and that keeping track of the normalization during training and back propagating through it (denoted norm=AP2 in the figure) performs even better and may be preferable to batch normalization in many scenarios such as recurrent NNs.

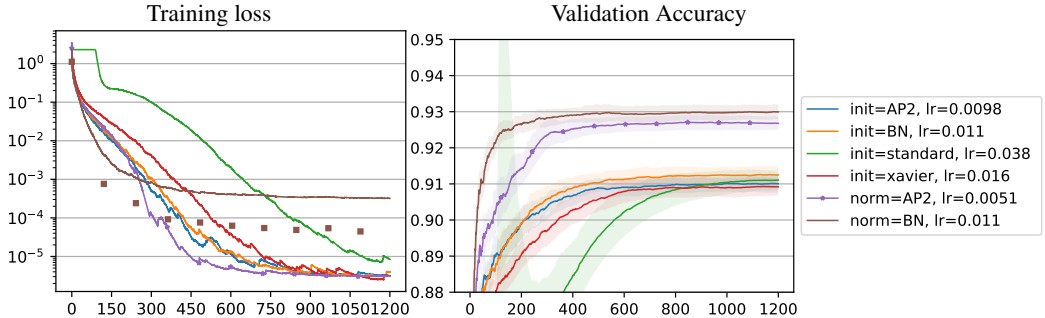

Figure C.2:   The effect of initialization/normalization on the progress of training. Observe that the initialization alone significantly influences the automatically chosen initial learning rate (lr) and the "trainability" of the network. Using the normalization during the training further improves performance for both batch and analytic normalization. BN has an additional regularization effect Ioffe (2017), the square markers in the left plot show BN training loss using averaged statistics.

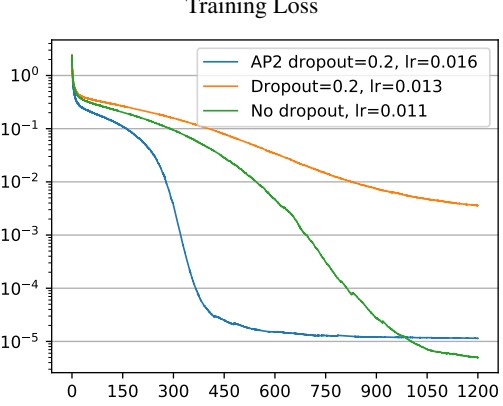

Figure C.3:   Training loss corresponding to Fig. 3. While stochastic dropout slows the training down due to increased stochasticity of the gradient, the analytic dropout smoothes the loss function and speeds the training up.

| | C | A | C | A | C | A | M | C | A | C | A | C | A | M | C | A | C | A | C | P | Softmax |
|---|---|---|---|---|---|---|---|---|---|---|---|---|---|---|---|---|---|---|---|---|---|
| $\sigma^*$ | 0 | 0.17 | 0.56 | 0.40 | 1.5 | 0.85 | 0.95 | 3.9 | 2.4 | 10 | 5.3 | 25 | 7.0 | 8.9 | 39 | 21 | 43 | 11 | 26 | 4.3 | |
| $\varepsilon_{\mu_1}$ | - | 0.00 | 0.00 | 0.03 | 0.21 | 0.05 | 0.21 | 0.96 | 0.11 | 0.43 | 0.11 | 0.71 | 0.09 | 0.18 | 0.79 | 0.14 | 0.37 | 0.10 | 0.26 | 0.97 | KL 0.06 |
| $\varepsilon_{\mu_2}$ | - | 0.00 | 0.00 | 0.01 | 0.01 | 0.01 | 0.09 | 0.42 | 0.05 | 0.22 | 0.04 | 0.19 | 0.03 | 0.11 | 0.59 | 0.07 | 0.18 | 0.08 | 0.19 | 0.73 | KL 0.03 |
| $\varepsilon_{\sigma_2}$ | - | 1.00 | 1.00 | 1.02 | 0.93 | 0.98 | 1.08 | 1.21 | 1.24 | 1.00 | 1.09 | 0.88 | 0.99 | 1.15 | 1.02 | 0.97 | 0.97 | 1.26 | 1.00 | 0.73 | |

Table C.1:   Accuracy of approximation of mean and variance statistics for each layer in a fully trained ConvPool-CNN-C network with dropout. A significant drop of accuracy is observed as well after max pooling, we believe due to the violation of the independence assumption.

## C.6   ACCURACY WITH MAX POOLING

Table C.1 shows accuracy of posterior approximation results for ConvPool-CNN-C, discussed above which includes max pooling layers. The network is trained and evaluated on CIFAR-10 with dropout the same way as in § 5.1.

