# OpenReview forum: "Feed-forward Propagation in Probabilistic Neural Networks with Categorical and Max Layers"
_ICLR.cc/2019/Conference_

### Official Review · AnonReviewer3 · 2018-10-25
**valid technical contribution. a little on the incremental side**

**Rating:** 6
**Confidence:** 4

**Review:**

The main contribution of the paper are methods for propagating approximate uncertainty in neural networks through max and argmax layers. The proposed methods are explained well. The paper is clearly written. The methods are validated in small scale experiments and seem to work well.

The proposed approach is not much more accurate than Monte Carlo dropout, but is more computationally efficient. The standard way of efficiently predicting at test time with a dropout-trained network is to simply scale the weights. Could the authors try calibration on networks of this type and compare against the proposed method with calibration? (i.e. scale the predicted logits of the standard test-time network to be on the same scale as the logits under your approach)

---

> ### Author Response · Authors · 2018-11-07
> **rebuttal and clarification requests**
>
> “Scaling weights in standard test-time dropout”:
> The method AP1 is consistent with the mentioned standard scaling (by the probability of the activation being not dropped). More precisely, we follow the common implementation of dropout (described in Srivastava et al., 2014.) where the multiplicative noise variable Z attains value $1/p$ with probability $p$ and $0$ with probability 1-p at training time. Since the expectation of this variable is 1, there is no scaling of weights needed at test time.
>
> “Calibration of AP1”
> This might be a valid idea, however it is not completely clear what is ment. Note that with AP2 we are scaling only the variances of the logits not logits themselves. In AP1 variances of logits are not available. The scaling of logits is a free degree of freedom of the last linear layer. Is the proposition then to estimate the final rescaling to maximize the likelihood of the validation set?

---

> > ### Comment · AnonReviewer3 · 2018-11-26
> > **yes**
> >
> > Yes I think estimating the final scaling of logits (by maximizing validation log likelihood) would provide a fair comparison. If your method would then still outperform it would be more convincing to me.

---

> > > ### Author Response · Authors · 2018-11-26
> > > **Yes, we agree**
> > >
> > > So far, we did the following:
> > > The variance calibration can be performed on the training set instead of the validation set. In the calibration of AP2 we correct the ratio of the noise due to dropout in the last layer. Essentially dropout samples are needed with any relevant data. Training data works just as fine here. We get same calibration ratios (two significant digits) and same test likelihoods as reported.
> > > For a fair comparison with AP1 method approximating standard dropout we estimate a constant variance to be added in the last layer. It is estimated as the average variance of samples due to dropout. We would not expect such calibration to be better than MC with many samples, which it tries to approximate. It does indeed improve test likelihoods of AP1 to the level of about MC-10 likelihood.
> > > These changes will be incorporated.
> > >
> > > Calibration of the scale parameter on the validation set is a totally different matter. All the considered methods may and would benefit from such cross-validation. Note also that results reported in the literature (eg. SOTA results we quote in Table 3) would be likely improved this way as well. We therefore prefer to leave this question out of scope of this contribution. Preliminary, cross-validation improves all methods in our experiments, making the test likelihoods not so clearly distinct. This also indicates that we should apply scale regularization during training to learn such models properly.

---

### Official Review · AnonReviewer1 · 2018-10-31

**Rating:** 6
**Confidence:** 5

**Review:**

This paper revisits the feed-forward propagation of mean and variance in neurons. In particular, it addresses the problem of propagating uncertainty through max-pooling layers and softmax. This is important since previous methods on probabilistic neural networks have not handled these challenges, hence preventing them from using max-pooling and softmax in a principled way.

In general, the authors did a good job approximating the mean and variance for the output of max-pooling and softmax. I have several concerns:

The authors claimed that they derived new approximation for leaky ReLU as well. It seems the approximation in Eq. (22)-(25) is exactly the same as Gast and Roth, 2018, both leveraging the results on obtaining the maximum of two Gaussian random variables.

The Bayesian formulation is not clear enough and seems a bit problematic in Sec. 2. For example, in Eq. (2), the authors mentioned p(X^k | x_0) as the posterior distribution. In this case, what is the corresponding prior? Besides, it should be made clear from the beginning that the network parameters W is not treated as random variables.

It is an interesting idea to incorporate the Gumbel distribution’s variance into the approximation in Eq. (10). Do you have any empirical results on how accurate the approximation in Eq. (10) is?

Similarly, the approximation from Eq. (13) to Eq. (14)-(15) seems a bit ad-hoc. It is good to know that the approximation is exact in the case of two input variables. However, it would be more convincing if the authors could investigate more about the accuracy of the approximation (either empirically or theoretically) when there are more than two variables.

The organization of the paper could be improved. The notion of nonlinearity is not mentioned until Sec. 3. When reading Sec. 2, one would wonder where the nonlinear transformation happens. It would help to clarify a bit at the start of Sec. 2.

In terms of experiments, one important benefit of feed-forward propagation is that it avoid the multi-pass MC estimates. However, it seems the performance boost on NLL mainly comes from the calibration, where \sigma^* needs to be computed using multi-pass MC estimates.

The noise level (std of 10^-4 and 0.01) seems quite small in Table 1. According to the results, it seems the error of \sigma_2 increases a lot as the noise level goes from 10^-4 to 0.01, suggesting that the approximation does not work well when the input noise is large. How is the accuracy when the noise level further increases?

Unlike the natural-parameter networks (NPN) in Wang et al. (2016), the proposed work assumes zero variance in the parameters W. It would be interesting to see whether the proposed methods could also improve NPN.

---

> ### Author Response · Authors · 2018-11-07
> **rebuttal and clarification requests**
>
> “Eq. (22)-(25) is exactly the same as Gast and Roth, 2018”
> Expressions for the mean are indeed the same and well known. Expressions for the variance are different.
> Our main observation here is that that the variance of the output is proportional to the variance of the input and the proportionality factor is a function of one variable $a$ that can be well approximated, guaranteeing non-negativity and correct asymptotes for large and small inputs. In contrast expression (13b) in [Gast and Roth] requires computing (10b) twice that involves a difference of expressions depending on normal cdf, does not readily simplify and may result in negative values. They mention of adding 1e-4 variance to all activations for numerical reason, a quite large value that suggests that the problem is not void. We do not need to add any such constant, furthermore, it could negatively impact the accuracy when all activations are small due to the scaling. This is particularly important in approximating dataset statistics (Fig B.1).
>
> “Posterior distribution. What is the corresponding prior?”
> The term posterior is used to denote the distribution of interest when conditioned on some observations, following graphical models (c.f. maximum a posteriori solution in MRF / CRF). (Technically, the a priori distribution of the outputs without any observations is also existing but in our case is not relevant).
> The weights are indeed not treated as random variables in this work. Were they random, we would still speak of the posterior distribution of the outputs given the inputs and of the posterior distribution of the weights given both inputs and outputs in the context of Bayesian learning.
>
> “W is not treated as random variables”
> This will be made clear. However, this is only for reasons of simplicity. W can be made random, in which case the variance for the linear layer needs to use the expression of the variance of product of independent random variables (weights and layer inputs). The propagation method however cannot be expected to work very well in convolutional networks since outputs will be strongly spatially correlated. This restriction applies to all related methods: Wang et al. (2016), Hern´andez-Lobato & Adams, 2015), etc. which do not do experiments with conv networks.
>
> “Empirical results on how accurate the approximation in Eq. (10) is”
> We interpret: “how accurate is the step to forget the shape of the Gumbel distribution and only take into account its variance?”. This is not exactly what happens, it is not a standalone approximation step. The approximating family for U with U_k = (X_y+Gamma_y) - (X_k+Gamma_k) is chosen to be multivariate logistic. This is the exact distribution in case X were deterministic (in case of two variables Gamma_2 - Gamma_1 is logistic). For random X, we lose in approximating the real distribution of U (which is hoped to be bell-shaped due to summing multiple terms) by a bell shaped multivariate logistic distribution. The step around (10) cannot be evaluated separately.
>
> “investigate more about the accuracy of the approximation (13)-(15)”
> This we will gladly do. We will generate distributions for X and evaluate the quality of both approximations w.r.t. MC sampling.
>
> “\sigma^* needs to be computed using multi-pass MC estimates”
> Only once after training is done. I.e. not adding to the cost of training iterations nor at the test time. Future work will address calibration during training.
>
> “The noise level (std of 10^-4 and 0.01) seems quite small in Table 1”
> The numbers $10^-4$ and $0.01$ are the variances. The standard deviations are thus 0.01 and 0.1, respectively. The input range is in the interval [0,1], so the noise level is not small. The accuracy can be compared to the results of
>
> Adel Bibi, Modar Alfadly, Bernard  Ghanem, "Analytic Expressions for Probabilistic Moments of PL-DNN with Gaussian Input", (CVPR 2018) [Oral]
> In Table 1, they evaluate the accuracy for LeNet of the statistics of logits with input noise variance=1 and signal range [0,255], which corresponds to std of 0.0039 in our scale, smaller than we evaluate. The ratio of variances 0.4-0.6 appears worse than both tested cases in our Table 1.
>
> “Relation to natural-parameter networks (NPN) Wang et al. (2016),”
> NPNs are a similar propagation method to ours. They are more general in allowing approximation by a member of exponential family in each layer. In practice, for networks with real-valued weights, the only reasonable model for the outputs of a linear layer is the Normal distribution (an exponential family member with mean and variance as sufficient statistics and unrestricted domain). They have simple propagation rules only with exponential nonlinearities. Their results may benefit from our numerically stable approximation for ReLU and the new approximation for softmax. Random weights is a difference that is not essential for the method as discussed above.

---

> > ### Comment · AnonReviewer1 · 2018-12-05
> > **Thanks for the response**
> >
> > Thanks for your response. It addressed some of my concerns. However, I still have concerns on the lack of rigor when mentioning the term posterior distribution and the notation. The explanation/clarification provided does not convince me unfortunately. That said, I am still happy if the authors could be more careful when using the term (e.g., for p(X^k | x_0)) in the revised version.
> >
> > I am not sure that the statement ‘the propagation method however cannot be expected to work very well in convolutional networks since outputs will be strongly spatially correlated’ is correct. Convolutional layers are still linear layers, and propagation methods like NPN should be fairly easy to handle them (with some work on naturally extending the FC linear-layer propagation to its convolutional version).

---

### Official Review · AnonReviewer2 · 2018-10-31
**Novel contribution to propagate uncertainty across argmax/max operations. Some experiments are missing to show the real benefit of the method in practical scenarios.**

**Rating:** 6
**Confidence:** 3

**Review:**

* Summary

The authors focus on the problem of uncertainty propagation DNN. The authors claim two main contributions: they revisit the assumptions of the feed forward method (proposed by several authors as an inference method for BNNs based on ADF/EP) and proposed a new approximation for argmax/max based functions that allows to propagated the first two moments analytically.

* Comments:

The authors claim two main contributions: an analysis for the feed forward method (sections 2 and 3) previously proposed by several authors as an inference method for BNN based on ADF/EP, and a new method to propagate the uncertainty through argmax/max based operations (section 4).

Regarding the first contribution, I was expecting some new insights about the method that I did not find. I would suggest to focus on the second contribution and refactor this section as a background section. I would make it shorter, focusing on the representation of probabilities as latent variables trough a function, which is the important bit to understand the real contribution of the paper described in section 4. I would also remove some examples that do not seem critical to understand the rest of the paper and just increase its length.

The second contribution is quite novel. The authors propose a new approximation of argmax/max operations. The firstly proposed an approximation for argmax operations, e.g. latent variable view of the softmax, that avoids resorting to the normal cdf function that has numerical stability issues. Secondly, they suggest an approximation for max based operations, e.g. leaky relu, that again, does not depend on the gaussian cdf.

In the experimental section, the authors test:
a)	The accuracy of the proposed method approximating the posterior of the neurons
b)	End-to-end training benefits

In a) they use MC to collect the ground truth statistics and compare the proposed method (AP2) with a classical NN (AP1). The analysis is nice but I miss a comparison with other state-of-the-art methods. In particular, the authors claim that the novelty of their method compared to other feed-forward methods is that they can propagate the uncertainty through argmax/max operations analytically. They do not compare with these other feed forwards methods to show the benefit of this.  This is shown in the end-to-end training experiments; however, I would like to see a direct comparison with the classical paper (Hern´andez-Lobato & Adams, 2015). Finally, one of the justifications of the approximations that they propose is to avoid the numerical issues of the standard cdf. Have the authors compared with this, e.g. eq 18a, 18b? Using a robust implementation of the normal cdf/pdf function and further truncating them to avoid negative variances?

typo: Shortly before eq. 12, Should not S_{n-1} be defined as the softmax operation?

---

> ### Author Response · Authors · 2018-11-07
> **rebuttal and clarification requests**
>
> “Contribution 1: new insights about the method?”
> We mean: 1) a clear self-contained derivation, 2) the latent variable view of sigmoid that later extends to softmax, 3) the connection to standard NNs, 4) possibility to choose the approximating family at each layer in order to simplify the propagation. We believe this is useful and may help understanding by non-experts. Note that the frequently cited ADF is an incremental method for parameter estimation. Only example 3 (ReLU) is not directly used in the subsequent constructions. Do you definitely recommend to shorten this part? The numerical evaluation of accuracy has not been reported before with such methods.
>
> “Contribution 2: an approximation for argmax operations that avoids resorting to the normal cdf function that has numerical stability issues”
> There is likely to be a misunderstanding. The challenge of argmax is that it is a multivariate nonlinear function. There were no previously proposed analytic approximations using the normal cdf or not. Furthermore, evaluating the multivariate normal cdf is a hard computational problem.
>
> To support the utility of the results, let us mention one more paper we discovered that achieved improvements in speech recognition with the uncertainty propagation but explicitly mentions that the approximation for softmax was an unsolved problem and a significant limitation:
> Astudillo et al. (2014) “ACCOUNTING FOR THE RESIDUAL UNCERTAINTY OF MULTI-LAYER PERCEPTRON BASED FEATURES”
>
> “A direct comparison with the classical paper (Hern´andez-Lobato & Adams, 2015)”
> This work performs Bayesian learning, which we don’t do. Our work is related to the paragraph “Incorporating the likelihood factors”. They describe the case of ReLU with a numerical fix. Do you mean specifically comparing this approximation alone against ours? We do not see any other direct comparison applicable. The softmax is not considered there (they consider regression problems with fully connected 1-4 layers).
>
> “Comparison with a robust implementation of the normal cdf/pdf”
> According to our experiments, the bottleneck in the accuracy is currently due to the independence assumption. More accurate calculation of the variance as in (Hern´andez-Lobato & Adams, 2015) is possible, but is more computationally costly. Truncation in order to force non-negativity of variance is particularly undesirable. Because the accumulated scaling in a deep network can lead to all activations being either large or small, a valid asymptotic behaviour is required. This is particularly important for normalization (Fig. B1). We propose that in the context of NNs cheaper approximations with valid asymptotes are more practical.
>
> “Should not S_{n-1} be defined as the softmax operation”
> We quote the definition by Malik and Abraham. There is indeed relation S_{n-1}(u)  = softmax(0, u_1, … u_{n-1}). Think of the case with two variables X_1, X_2: we have U_1 =X_1 - X_2  and S_{1}(u) = sigmoid(u).

---

### Author Response · Authors · 2018-11-23
**Revision**

We thank all reviewers for their comments. We revisited the submission accordingly as follows.

A comprehensive evaluation of how well the proposed approximations models the simulated expectations as well as its derivatives in individual layers (LReLU, Softmax, Argmax) is included now in the Appendix B.

As was pointed out by Reviewer 1, the linear time approximation for softmax was ad-hoc. This has showed up in the evaluation, esp. for larger noises. We have revisited this approximation (see Linear Time Approximation in 4.1). It is still linear complexity and has a good accuracy (evaluated in Appendix B). This did not affect other results in the paper because the learned models have small variance on the output (a consequence of maximum likelihood estimation) for which the ad-hoc approximation was accurate enough.

The evaluation shows that the approximation for softmax is reasonable in a big range in the sense that it models a function that is very similar to the expected value of softmax and has similar gradients.  All proposed approximations are comparable to and consistently more accurate for small noise range than MC-100. When a higher accuracy is required (in some other potential applications) an MC estimate correcting our approximations can be used (i.e. using analytic approximation as a baseline for variance reduction).

We removed the example with ReLU and updated the presentation of the maximum of two variables and Leaky ReLU to cite the preceding work and highlight that we propose a simplification, which is more practical for use in end-to-end training (sec. 4.3, evaluated in in Appendix B).

We made the necessary clarifications (confusion about posterior, non-random weights, notation of the noise variance in the experiments, etc.).

More advanced calibration using the validation set (discussed with Reviewer 3) of AP1 and calibration of AP1 / AP2 during learning is left for future work.

---

### Meta-Review · Area_Chair1 · 2018-12-14

**Confidence:** 4
**Recommendation:** Accept (Poster)

**Metareview:**

Reviewers are in a consensus and recommended to accept after engaging with the authors. Please take reviewers' comments into consideration to improve your submission for the camera ready.